# The Impact of Tumor Stage and Histopathology on Survival Outcomes in Esophageal Cancer Patients over the Past Decade

**DOI:** 10.3390/medsci12040070

**Published:** 2024-12-09

**Authors:** Ayrton Bangolo, Vignesh Krishnan Nagesh, Grace Simonson, Abhishek Thapa, Arun Ram, Nithin Jayan Santhakumari, Rayan Chamroukh, Vivek Joseph Varughese, Shallot Nareeba, Aiswarya Menon, Kousik Sridharan, Angel Ann Chacko, Charlene Mansour, Daniel Elias, Gurinder R. Singh, Aaron Rambaransingh, Luis Roman Mendez, Charlotte Levy, Izage Kianifar Aguilar, Ibrahim Hamad, Urveesh Sharma, Jose Salcedo, Hadrian Hoang-Vu Tran, Abdullah Haq, Tahir B. Geleto, Kaysha Jean, Luis Periel, Sara Bravin, Simcha Weissman

**Affiliations:** 1Department of Hematology and Oncology, John Theurer Cancer Center, Hackensack University Medical Center, Hackensack, NJ 07601, USA; ayrtonbangolo@yahoo.com; 2Department of Internal Medicine, Hackensack Palisades Medical Center, North Bergen, NJ 07047, USAaiswaryamenon2795@gmail.com (A.M.); angelannc74@gmail.com (A.A.C.); cm1621@njms.rutgers.edu (C.M.); de217@njms.rutgers.edu (D.E.); izagekianifar@gmail.com (I.K.A.); luispape98@gmail.com (L.P.); simchaweissman@gmail.com (S.W.); 3Department of Internal Medicine, University Hospitals of Leicester NHS Trust, Leicester LE1 5WW, UK; js.nithin1994@gmail.com; 4Department of Internal Medicine, University of South Carolina, Columbia, SC 29208, USA; vivekjvarughese@gmail.com

**Keywords:** esophageal cancer, mortality, SEER, interaction analysis, adenocarcinoma, squamous cell carcinoma

## Abstract

Background: Esophageal cancer (EC) is the sixth leading cause of cancer-related mortality worldwide, continuing to be a significant public health concern. The purpose of this study is to assess the impact of staging and histopathology of EC on associated mortality. The study also aims to further investigate clinical characteristics, prognostic factors, and survival outcomes in patients diagnosed with EC between 2010 and 2017. Furthermore, we analyzed the interaction between tumor histology and staging and the risk of mortality. Methods: A total of 24,011 patients diagnosed with EC between 2010 and 2017 in the United States were enrolled from the Surveillance, Epidemiology, and End Results (SEER) database. Demographic parameters, tumor stage, and histologic subtypes were analyzed and associated overall mortality (OM) and cancer-specific mortality (CSM) were measured across all subgroups. Covariates reaching the level of statistical significance, demonstrable by a *p*-value equal to or less than 0.01, were incorporated into a multivariate Cox proportional hazards model. A hazard ratio greater than 1 was indicative of an increased risk of mortality in the presence of the variable under discussion. Additionally, the study explores the interaction between histology and tumor stage on outcomes. Results: The majority of patients were male (80.13%) and non-Hispanic white (77.87%), with a predominant age at diagnosis of between 60 and 79 years (59.86%). Adenocarcinoma was the most common tumor subtype (68.17%), and most patients were diagnosed at a distant stage (41.29%). Multivariate analysis revealed higher mortality risks for males, older patients, unmarried individuals, and those with advanced-stage tumors. Higher income, receiving radiation or chemotherapy, and undergoing surgery were associated with lower mortality. Tumor subtype significantly influenced mortality, with squamous cell carcinoma and neuroendocrine tumors showing higher hazard ratios compared to adenocarcinoma. Adenocarcinoma is linked to a poorer prognosis at advanced stages, whereas the opposite trend is observed for SCC. Conclusions: The study identifies significant demographic and clinicopathologic factors influencing mortality in esophageal cancer patients, highlighting the importance of early diagnosis and treatment intervention. Future research should focus on tailored treatment strategies to improve survival outcomes in high-risk groups and to understand the interaction between tumor histology and tumor stage.

## 1. Introduction

Esophageal carcinoma is one of the most prevalent cancers all over the world with a prevalence of over 450,000 cases globally [1]. The two most common histological subtypes of esophageal cancers are squamous cell carcinoma and adenocarcinoma, with incidences varying globally and in Western countries. Squamous cell carcinoma constituted the majority of esophageal cancers globally and in Western countries in the 20th century [2]. However, the rates of adenocarcinoma are increasing rapidly in western countries, leading to a recent change in the incidence patterns, while the trend has stayed the same in developing countries with SCC being the predominant histology type [2,3]. Incidences of adenocarcinoma in the United States have been increasing steadily compared to squamous cell carcinoma, with the trends of incidence in the years 1998–2003 showing that squamous cell carcinoma incidence fell by 3.7% year-on-year and adenocarcinoma incidence increased by 2.7% per year, according to data collected by the National Program of Cancer Registries or the Surveillance, Epidemiology, and End Results program [4]. An increase in the incidence of adenocarcinoma in recent years is clearly visible. In 1973 the incidence of adenocarcinoma was 0.5 per 100,000 white males per year, which increased to 3.2 per 100,000 in the year 1997, and again showed a 20.7% increase in the year 2000 with an estimated 3.7 per 100,000 [2]; whereas that of adenocarcinoma in the United States continued to rise, with it being 5.31 per 100,000 person years in the period 2003–2007 according to the Connecticut tumor registry [5].

The risk factors vary according to histology—Barrett’s esophagus and obesity are two major risk factors for adenocarcinoma of the esophagus, with Barrett’s esophagus being a known pre-malignant condition [6,7]. The major risk factors for squamous cell carcinoma include male gender, smoking, alcohol consumption, dietary factors, and family history [6]. The typical symptoms of esophageal cancer include heartburn, retrosternal pain, sensation of pressure, and dysphagia with advanced cases showing anemia, bleeding, and a loss of weight and appetite [8]. Occasionally, chronic gastrointestinal bleeding leading to anemia can be the presenting complaint of esophageal cancer [9]. The diagnosis of esophageal cancer is usually through an endoscopic biopsy followed by histological diagnosis [10]. Most cases of esophageal cancers are diagnosed after the tumor is locally advanced, causing symptoms such as dysphagia due to decreased lumen size, as evidenced by the fact that only one in eight cases are diagnosed at stage T1 [10,11]. The different modalities of treatment for esophageal cancer vary according to histology, stage of tumor, and the location of the tumor. These range from endoscopic resection for superficial tumors to an esophagectomy followed by postoperative radiation with or without chemotherapy, with post-operative chemo radiation showing improved outcomes in patients [12,13].

The impact of different subtypes of EC on mortality is still controversial in the literature. Some studies have indicated that adenocarcinoma is associated with higher mortality, while others have suggested that squamous cell carcinoma (SCC) carries a greater risk. This study aims to explore the various prognostic factors, including histopathology, influencing mortality in esophageal cancer, with a specific focus on the interaction between age and tumor subtype to clarify these discrepancies.

## 2. Methods

This population-based retrospective cohort study enrolled patients with esophageal cancer (EC) from the SEER Research Plus database, based on 18 population registries, accessed on 21 June 2024. SEER, sponsored by the U.S. National Cancer Institute (NCI), is a leading source of cancer-related data, covering 28% of the U.S. population. Histological and topographical codes were used to identify EC cases, while excluding patients with unknown age, race, or cancer stage.

Sociodemographic and clinicopathological variables, such as age, gender, race, tumor site, cancer stage, income, and treatments (surgery, radiation, or chemotherapy), were analyzed. The primary outcomes were overall mortality (OM), representing deaths from any cause, and cancer-specific mortality (CSM), referring to deaths from EC-related complications.

A Cox proportional hazard regression model was applied, with variables showing *p* < 0.01 in univariate analysis being included in multivariate analysis. A hazard ratio (HR) > 1 indicated negative prognostic factors. Statistical significance was set at *p* < 0.01, and the analysis was performed using STATA 18 software.

The analysis included 24,011 patients diagnosed with EC between 2010 and 2017. As demonstrated in Table 1, this cohort predominantly represented males (80.13%), individuals between the ages of 60 and 79 at the time of diagnosis (59.86%), married individuals (55.44%), adenocarcinoma tumor subtype (68.17%), cancer diagnosed at the distant stage (41.29%), non-Hispanic whites (77.87%), residents living in counties in metropolitan areas of 1 million people (53.93%), and those earning an annual income of USD 75,000 or more (34.37%). At the time of diagnosis, the dispersion of age consisted of 0.99% of patients aged 0–39, 24.00% aged 40–59, 59.86% aged 60–79, and 15.15% aged 80 or older. Distribution across marital status revealed that 55.44% were married, 22.33% were single, 11.92% were divorced or separated, and 10.32% were widowed. Tumor subtype analysis showed that the majority of cancers were adenocarcinoma at 68.17%, followed by squamous cell carcinoma at 30.29%, adenosquamous tumors at 0.71%, neuroendocrine tumors at 0.69%, and gastrointestinal stromal tumors (GIST) at 0.14%. Analysis of tumor staging showed a majority of tumors at the distant stage (41.29%), with others at the locally advanced (37.03%) and localized (21.68%) stages. Non-Hispanic whites comprised the majority of the racial demographic at 77.87%, with non-Hispanic blacks comprising 8.73%, Hispanics comprising 7.94%, and the remaining 5.45% accounted for by others. The analysis of treatment modalities utilized by patients in the cohort revealed that 57.82% received radiation, 65.57% received chemotherapy, and 26.42% received surgery.

Table 2 presents the crude analysis of various factors associated with all-cause mortality and cancer-related mortality among US patients diagnosed with esophageal cancer between 2010 and 2017. Patients who were male saw significantly increased hazard ratios of 1.06 (95% CI 1.02–1.10, *p* < 0.01) for overall mortality and 1.07 (95% CI 1.03–1.11, *p* < 0.01) for cancer-related mortality relative to female patients. Age at diagnosis showed statistical significance with greater mortality hazards among older individuals, increasing to a hazard ratio of 1.75 (95% CI 1.50–2.04, *p* < 0.01) for overall mortality and 1.47 (95% CI 1.25–1.72, *p* < 0.01) for cancer-related mortality among individuals aged 80 years or older. Relative to married individuals, increased risks were observed among single individuals (HR = 1.17, 95% CI 1.13–1.21, *p* < 0.01) for overall mortality and (HR = 1.16, CI 95% 1.11–1.20, *p* < 0.01) cancer-related mortality; divorced/separated individuals (HR = 1.22, 95% CI 1.17–1.28, *p* < 0.01) for overall mortality and (HR = 1.24, 95% CI 1.18–1.30, *p* < 0.01) cancer-related mortality; and widowed individuals (HR = 1.39, 95% CI 1.33–1.46, *p* < 0.01) for overall mortality and (HR = 1.33, 95% CI 1.26–1.40, *p* < 0.01) cancer-related mortality.

Hazard ratios differed significantly among tumor subtypes relative to adenocarcinoma. The greatest increase was seen in neuroendocrine tumors (NE) (HR = 1.76, 95% CI 1.49–2.08, *p* < 0.01) for overall mortality and (HR = 1.70, 95% CI 1.41–2.03, *p* < 0.01) cancer-related mortality. The greatest decrease was seen in gastrointestinal stromal tumors (GIST) (HR = 0.26, 95% CI 0.15–0.46, *p* < 0.01) for overall mortality and (HR = 0.16, 95% CI 0.07–0.36, *p* < 0.01) cancer-related mortality. Tumor stage was associated with increased hazard ratios in advanced stages, progressing to 3.19 (95% CI 3.06–3.32, *p* < 0.01) for overall mortality and 4.13 (95% CI 3.94–4.34, *p* < 0.01) for cancer-related mortality in the distant stage category.

Increased risks were observed among non-Hispanic black individuals (HR = 1.30, 95% CI 1.24–1.37, *p* < 0.01) for overall mortality and (HR = 1.31, 95% CI 1.24–1.38, *p* < 0.01) cancer-related mortality as well as among Hispanic individuals (HR = 1.09, 95% CI 1.03–1.15, *p* < 0.01) for overall mortality and (HR = 1.11, 95% CI 1.05–1.18, *p* < 0.01) cancer-related mortality. Increased hazard ratios were observed among individuals living in counties in metropolitan areas of 250,000 people (HR = 1.12, 95% CI 1.06–1.17, *p* < 0.01) for overall mortality and (HR = 1.13, 95% CI 1.07–1.19, *p* < 0.01) for cancer-related mortality. Similarly, increased hazard ratios were observed among those living in nonmetropolitan counties adjacent to a metropolitan area (HR = 1.07, 95% CI 1.01–1.13, *p* < 0.05) and those living in nonmetropolitan counties not adjacent to a metropolitan area (1.09, 95% CI 1.02–1.17, *p* < 0.05). Greater annual income showed decreased hazard ratios, with the lowest risks seen among those earning $75,000 or more (HR = 0.76, 95% CI 0.67–0.87, *p* < 0.01) in overall mortality and (HR = 0.80, 95% CI 0.69–0.92, *p* < 0.01) in cancer-related mortality. Patients who received radiation, chemotherapy, or surgical treatment all showed significantly decreased hazard ratios, with the greatest decrease seen in those who received surgery (HR = 0.30, 95% CI 0.29–0.31, *p* < 0.01) for overall mortality and (HR = 0.27, 95% CI 0.25–0.28, *p* < 0.01) cancer-related mortality. These results demonstrate the significant impact of demographic, socioeconomic, and clinicopathologic factors on mortality outcomes among individuals diagnosed with esophageal cancer.

Table 3 presents the results of multivariate Cox proportional hazard regression analyses of factors that influenced overall and cancer-related mortality among patients living in the United States who were diagnosed with esophageal cancer between 2010 and 2017. Male patients were at a higher risk with increased adjusted proportional hazard ratios when compared to female patients (HR = 1.15, 95% CI 1.11–1.20, *p* < 0.01) for overall mortality and (HR = 1.13, 95% CI 1.08–1.18, *p* < 0.01) cancer-related mortality. At the time of diagnosis, hazard ratios increased with increasing patient age for both all-cause and cancer-related mortality. The highest risk was seen in the 80 or older category (HR = 1.62, 95% CI 1.39–1.90, *p* < 0.01) for all-cause and (HR = 1.40, 95% CI 1.20–1.65, *p* < 0.01) cancer-related mortality. Marital status was found to significantly influence hazard ratios. For all-cause mortality, risk progressed from a hazard ratio of 1.11 (95% CI 1.07–1.15, *p* < 0.01) in single individuals, to a hazard ratio of 1.16 (95% CI 1.10–1.22, *p* < 0.01) in widowed individuals, to the highest hazard ratio of 1.19 (95% CI 1.14–1.25, *p* < 0.01) in divorced/separated individuals. This trend was likewise observed in cancer-related mortality, with a hazard ratio of 1.09 (95% CI 1.04–1.13, *p* < 0.01) in single individuals, 1.13 (95% CI 1.07–1.20, *p* < 0.01) in widowed individuals, and divorced/separated individuals at the highest risk with a hazard ratio of 1.20 (95% CI 1.14–1.26, *p* < 0.01).

Tumor subtypes exhibited a significant influence on hazard ratios when compared to adenocarcinoma. For all-cause mortality, hazard ratios increased in patients with squamous cell carcinoma to 1.38 (95% CI 1.27–1.49, *p* < 0.01) and adenosquamous tumors to 2.06 (95% CI 1.24–3.42, *p* < 0.01). For cancer-related mortality, similar increases were seen in patients with squamous cell carcinoma (HR = 1.45, 95% CI 1.33–1.59, *p* < 0.01) and adenosquamous tumors (HR = 2.41, 95% CI 1.37–4.26, *p* < 0.01), in addition to those with neuroendocrine tumors (HR = 2.12, 95% CI 1.17–3.85, *p* < 0.05). Gastrointestinal stromal tumors (GIST) were the only tumor subtype in which hazard ratios decreased, in terms of both all-cause mortality (HR = 0.29, 95% CI 0.13–0.65, *p* < 0.01) and cancer-related mortality (HR = 0.07, 95% CI 0.01–0.52, *p* < 0.01) conditions. Notably, non-Hispanic blacks were the only demographic to exhibit a markedly heightened mortality, with hazard ratios of 1.09 (95% CI 1.03–1.15, *p* < 0.01) overall and 1.10 (95% CI 1.03–1.16, *p* < 0.01) in cancer-related mortality. Annual income emerged as a factor affecting the adjusted proportional hazard ratio only in the category of USD 75,000 or more (HR = 0.85, 95% CI 0.73–0.98, *p* < 0.05) for overall mortality.

All three treatment modalities analyzed in this cohort improved patient outcomes. Relative to patients who did not undergo treatment, for overall mortality, hazard ratios decreased to 0.95 (95% CI 0.92–0.98, *p* < 0.01) in patients who underwent radiation, 0.52 (95% CI 0.50–0.54, *p* < 0.01) in those who underwent chemotherapy, and 0.37 (95% CI 0.36–0.39, *p* < 0.01) in those who underwent surgery. Outcomes were nearly identical for cancer-related mortality, with hazard ratios falling to 0.94 (95% CI 0.91–0.98, *p* < 0.01) in those who underwent radiation, 0.51 (95% CI 0.49–0.53, *p* < 0.01) in those who underwent chemotherapy, and 0.35 (95% CI 0.34–0.37, *p* < 0.01) in those who underwent surgery compared to patients who did not receive treatment. Living area did not show any significant impact on health outcomes in this study. These multivariate analyses reflect the considerable influence of demographic, socioeconomic, and clinicopathologic factors on all-cause and cancer-related mortality in patients diagnosed with esophageal cancer.

Table 4 presents the results from multivariate Cox proportional hazard regression analyses of factors affecting overall mortality (OM) and cancer-specific mortality (CSM) in U.S. patients diagnosed with esophageal cancer between 2010 and 2017, considering interactions between tumor subtype and stage.

Adenocarcinoma (AC) at the localized stage was used as the reference group. Patients with locally advanced AC had significantly increased hazard ratios for both OM (HR = 2.15, 95% CI 2.03–2.28, *p* < 0.01) and CSM (HR = 2.70, 95% CI 2.52–2.88, *p* < 0.01). Similarly, patients with distant AC also showed notably elevated HRs for OM (HR = 3.35, 95% CI 3.17–3.55, *p* < 0.01) and CSM (HR = 4.36, 95% CI 4.08–4.66, *p* < 0.01).

For squamous cell carcinoma (SCC), the hazard ratios for OM and CSM were significantly lower than those for AC. Locally advanced SCC was associated with a decreased risk of OM (HR = 0.64, 95% CI 0.58–0.70, *p* < 0.01) and CSM (HR = 0.57, 95% CI 0.51–0.63, *p* < 0.01). Similarly, distant SCC was associated with a reduced risk for both OM (HR = 0.73, 95% CI 0.67–0.80, *p* < 0.01) and CSM (HR = 0.69, 95% CI 0.62–0.76, *p* < 0.01).

Neuroendocrine (NE) tumors at the locally advanced stage exhibited a non-significant HR for OM and a reduced HR for CSM (HR = 0.43, 95% CI 0.22–0.87, *p* < 0.05). The HRs of NE tumors at distant stages were non-significant for OM and CSM.

In patients with adenosquamous carcinoma (ASC), the locally advanced stages had HRs that were non-significant for OM and CSM. The HR for ASC tumors at distant stages was non-significant for OM and showed modest reductions for CSM (HR = 0.52, 95% CI 0.28–0.96, *p* < 0.05).

Lastly, gastrointestinal stromal tumors (GIST) at the locally advanced stage exhibited a non-significant HR for OM and a highly elevated risk for CSM (HR = 14.47, 95% CI 1.31–159.87, *p* < 0.05). The HRs for GIST tumors at distant stages were non-significant for OM and CSM.

In summary, tumor stage and subtype had a marked impact on survival outcomes, with AC at advanced stages and GIST at the locally advanced stage being associated with significantly worse survival outcomes. Conversely, SCC and NE tumors showed more favorable prognoses.

## 3. Discussion

This analysis involved 24,011 patients diagnosed with esophageal cancer (EC) between 2010 and 2017. The cohort was predominantly male (80.13%), with most patients aged 60–79 (59.86%). The majority were married (55.44%) and had been diagnosed with adenocarcinoma (68.17%). Most cancers were identified at a distant stage (41.29%), and a significant portion of the cohort were non-Hispanic whites (77.87%). Regarding treatment, 57.82% received radiation, 65.57% chemotherapy, and 26.42% surgery. Additionally, 53.93% lived in metropolitan areas with over 1 million people, and 34.37% had annual incomes of USD 75,000 or more. Key findings showed that male patients, older age groups (especially those 80 or older), and individuals who were divorced or separated had higher mortality risks. Tumor subtypes also influenced outcomes, with squamous cell carcinoma and adenosquamous carcinoma associated with higher mortality compared to adenocarcinoma, and gastrointestinal stromal tumors (GIST) showing notably lower hazard ratios. Additionally, treatment modalities such as surgery, chemotherapy, and radiation significantly reduced mortality risks. Interactions between tumor stage and subtype revealed that adenocarcinoma in advanced stages exhibited worse outcomes, while squamous cell carcinoma and neuroendocrine tumors showed more favorable survival outcomes, especially in locally advanced stages.

The predominance of males (80.13%) aligns with previous studies that have established esophageal cancer as more common in men than women, likely due to lifestyle factors such as tobacco use, alcohol consumption, and obesity [14,15]. The majority of patients being aged 60–79 (59.86%) is consistent with the understanding that EC is primarily a disease of the elderly, further emphasizing the need for vigilant screening and early diagnosis in older populations [16,17]. The high prevalence of adenocarcinoma (68.17%) among this cohort is noteworthy, as it confirms the rising incidence of esophageal adenocarcinoma, particularly in Western countries, where the increase in obesity and gastroesophageal reflux disease (GERD) are contributing factors leading to Barrett’s esophagus [18,19,20]. The majority of cancers being diagnosed at a distant stage (41.29%) indicates a concerning trend in late-stage diagnoses, owing to the non-specific symptoms and asymptomatic nature in early stages [21].

Treatment modalities reveal that a significant proportion of patients received chemotherapy (65.57%), radiation (57.82%), or surgery (26.42%), reflecting current clinical practice, where multimodal treatment is often employed in advanced stages of esophageal cancer [22,23,24]. Surgery remains a key treatment option, especially in localized and locally advanced stages, but its lower usage rate may reflect the high number of late-stage diagnoses where surgery is not feasible [21].

Socioeconomic factors, such as the observation that 34.37% of patients had an annual income of USD 75,000 or more, may also play a role in treatment accessibility and outcomes, as higher income is often associated with better access to healthcare services. The finding that 53.93% of the cohort lived in metropolitan areas further supports the notion that geographic location can impact the timeliness and quality of cancer care [25].

Male patients were found to be at a higher risk of both overall and cancer-related mortality compared to females. This finding aligns with the well-established higher incidence of EC in males due to lifestyle factors such as higher tobacco and alcohol use [26,27]. Increasing age was another significant predictor, with the highest risk observed in patients aged 80 or older, consistent with previous studies that show age is a critical determinant of survival in EC likely due to having reduced physiological reserves required to tolerate surgery and the increased likelihood of comorbidities in older populations, which complicates treatment and recovery [28]. Marital status showed a notable influence on mortality, with the highest hazard ratios in divorced or separated individuals, followed by widowed and single patients as in the case of majority of the tumors [29]. These results are consistent with prior research suggesting that social support provided by marriage can improve cancer outcomes by increasing adherence to treatment and follow-up care [30]. Tumor subtype played a crucial role, with squamous cell carcinoma and adenosquamous tumors exhibiting significantly higher mortality risks compared to adenocarcinoma, which could be explained by the higher recurrence rates and presence of other secondary primary tumors [31]. Studies have shown that patients with adenocarcinoma have a better likelihood of long-term survival after an esophagectomy compared to SCC [32]. Patients with neuroendocrine tumors of the esophagus had a higher risk of mortality, as supported by current literature, likely due to diagnosis at advanced stages owing to the aggressive nature of the tumor [33]. Patients with adenosquamous carcinoma of the esophagus had a higher risk of mortality compared to adenocarcinoma, which is similar to the results of the study conducted by Chen et al.; however, the sample size of ASC was small in our study [34]. Notably, patients with gastrointestinal stromal tumors (GIST) had significantly lower hazard ratios for both overall and cancer-related mortality, potentially reflecting the improved outcomes with targeted therapies such as tyrosine kinase inhibitors in GIST [35]. However, the sample size of patients with GIST was negligible and the data on mortality of patients with GIST compared to other esophageal cancer subtypes are limited due to its rarity.

Non-Hispanic black patients demonstrated slightly higher mortality rates compared to other racial groups, consistent with previous studies indicating racial disparities in cancer outcomes due to factors like differences in healthcare access, and present at advanced stages at diagnosis with a lower surgical rate in NH blacks compared to whites [36,37]. Some studies have shown that non-Hispanic blacks have a higher incidence of SCC, which is associated with higher mortality [38]. Interestingly, annual income also impacted mortality risk, with patients earning USD 75,000 or more showing a decreased hazard ratio. This suggests that higher income may facilitate better access to early diagnosis and advanced treatment options, improving survival outcomes [38]. Treatment was strongly associated with improved survival. Patients undergoing surgery, chemotherapy, or radiation had significantly lower mortality rates compared to those who did not receive treatment [39,40]. This is consistent with previous evidence supporting the role of multimodal therapy in improving survival in patients with locally advanced or resectable EC. Geographic location (i.e., living in metropolitan areas) did not significantly impact outcomes in this study, which contrasts with prior findings suggesting that rural populations often have worse cancer outcomes due to limited access to specialized care [41].

The unique findings from this study highlight the crucial impact of tumor subtype and stage on survival outcomes in patients diagnosed with esophageal cancer. Adenocarcinoma at advanced stages was associated with significantly worse survival, as evidenced by markedly higher hazard ratios for both overall mortality and cancer-specific mortality. Patients with locally advanced AC had a 2.15-fold increase in OM risk and a 2.70-fold increase in CSM risk, while those with distant AC exhibited even higher risks, with hazard ratios of 3.35 for OM and 4.36 for CSM. These results are consistent with previous research indicating that advanced-stage adenocarcinoma carries a particularly poor prognosis due to its aggressive nature and often delayed diagnosis [42].

In contrast, patients with squamous cell carcinoma (SCC) displayed significantly lower hazard ratios for both OM and CSM compared to those with AC at advanced stages, even though SCC was associated with higher mortality compared to AC, as mentioned above. These findings suggest that SCC, despite being associated with poorer outcomes in other studies, may confer a relative survival advantage compared to AC in certain stages. This could be likely due to the fact that the sample size of patients with advanced SCC is small compared to AC.

The emergence of artificial intelligence has been advantageous in the detection of esophageal carcinoma (EC). Studies have explored the use of narrow-band imaging (NBI) in conjunction with the You Only Look Once (YOLO) framework to enhance early detection, showing improved accuracy and recall for dysplasia compared to white light imaging (WLI). Results indicate that the YOLOv5 model, which employs hyperspectral narrowband images, significantly surpassed YOLOv8 in identifying squamous cell carcinoma (SCC) and dysplasia, achieving precision rates of 85.1% and 81.7%, respectively, while highlighting the necessity for larger datasets to further enhance the model’s performance [43,44,45].

Neuroendocrine (NE) tumors and adenosquamous carcinoma (ASC) had more variable outcomes. NE tumors at distant stages did not show significant hazard ratios, but locally advanced NE tumors were associated with a reduced risk of CSM. For ASC, hazard ratios for OM and CSM at advanced stages were generally non-significant, although distant ASC showed a modest reduction in CSM risk. The small sample sizes for these tumor subtypes likely contribute to the variability in results, a limitation that has been noted in similar studies.

Our study analyzed a large cohort of 24,011 patients with a tissue-confirmed diagnosis of esophageal cancer, using strict inclusion and exclusion criteria to ensure a focused patient population.

## 4. Limitations

Data source limitations: The study relies on the SEER database, which may not capture all patients or reflect regional variations in treatment and outcomes.

Retrospective nature: As a retrospective analysis, the study is subject to biases related to the selection of patients and the completeness of medical records.

Lack of treatment details: While treatment modalities were analyzed, the SEER database does not provide detailed information on the specific treatments received by patients.

Comorbidities: The SEER database does not provide data on other comorbidities of patients.

Histological variability: Variations in histopathological assessments and reporting practices across institutions may introduce inconsistencies in the classification of tumor types.

## 5. Conclusions

This study emphasizes the critical impact of demographic, clinical, and histopathologic factors on mortality outcomes in esophageal cancer patients. Our findings reveal that male gender, older age, and advanced tumor stages are associated with increased mortality, while interventions such as surgery, radiation, and chemotherapy are linked to improved survival. Tumor histology, particularly the subtype of esophageal cancer, plays a significant role in influencing outcomes, with adenocarcinoma showing a distinct prognosis at different stages compared to squamous cell carcinoma and neuroendocrine tumors. These insights emphasize the need for early detection and personalized treatment approaches to enhance survival rates. Future research should delve deeper into optimizing treatment strategies for high-risk populations and further exploring the interplay between tumor histology and staging to refine therapeutic and prognostic models.

## Figures and Tables

**Table 1 medsci-12-00070-t001:** Demographic and clinicopathologic characteristics of US patients diagnosed with esophageal cancer between 2010 and 2017.

Characteristics		
Total	*n*	%
	**24,011**	**100**
**Gender**		
Female	4770	19.87
Male	19,241	80.13
**Age at diagnosis, y.o.**		
00–39	237	0.99
40–59	5763	24.00
60–79	14,373	59.86
80+	3638	15.15
**Marital status**		
Married	13,311	55.44
Single	5361	22.33
Divorced/separated	2862	11.92
Widowed	2477	10.32
**Tumor Subtype**		
Adenocarcinoma	16,369	68.17
SCC	7274	30.29
NE	165	0.69
Adenosquamous	170	0.71
GIST	33	0.14
**Tumor stage**		
Localized	5205	21.68
Locally advanced	8891	37.03
Distant	9915	41.29
**Race**		
Non-Hispanic white	18,698	77.87
Non-Hispanic black	2097	8.73
Hispanic	1907	7.94
Other	1309	5.45
**Living area**		
Counties in metropolitan areas of 1 million people	12,949	53.93
Counties in metropolitan areas of 250,000 to 1 million people	5458	22.73
Counties in metropolitan areas of 250,000 people	2133	8.88
Nonmetropolitan counties adjacent to a metropolitan area	2063	8.59
Nonmetropolitan counties not adjacent to a metropolitan area	1408	5.86
**Income per year**		
<$35,000	277	1.15
$35,000–44,999	1623	6.76
$45,000–54,999	2991	12.46
$55,000–64,999	4865	20.26
$65,000–74,999	6003	25.00
$75,000+	8252	34.37
**Radiation**		
No	10,127	42.18
Yes	13,884	57.82
**Chemotherapy**		
No	8268	34.43
Yes	15,473	65.57
**Surgery**		
No	17,667	73.58
Yes	6344	26.42
**Year of diagnosis**		
2010	2805	11.68
2011	2851	11.87
2012	2914	12.14
2013	2967	12.36
2014	3010	12.54
2015	3167	13.19
2016	3122	13.00
2017	3175	13.22

**Table 2 medsci-12-00070-t002:** Crude analysis of factors associated with all-cause mortality and cancer-related mortality among US patients diagnosed with esophageal cancer between 2010 and 2017.

Characteristics	Overall Mortality: Crude Proportional Hazard Ratio (95% Confidence Interval)	Cancer-Related Mortality: Crude Proportional Hazard Ratio (95% Confidence Interval)
**Gender**		
Female	1 (reference)	1 (reference)
Male	1.06 (1.02–1.10) **	1.07 (1.03–1.11) **
**Age at diagnosis, y.o.**		
00–39	1 (reference)	1 (reference)
40–59	1.14 (0.98–1.33)	1.09 (0.93–1.27)
60–79	1.16 (1.00–1.35)	1.03 (0.88–1.20)
80+	1.75 (1.50–2.04) **	1.47 (1.25–1.72) **
**Marital status**		
Married	1 (reference)	1 (reference)
Single	1.17 (1.13–1.21) **	1.16 (1.11–1.20) **
Divorced/separated	1.22 (1.17–1.28) **	1.24 (1.18–1.30) **
Widowed	1.39 (1.33–1.46) **	1.33 (1.26–1.40) **
**Tumor Subtype**		
Adenocarcinoma	1 (reference)	1 (reference)
SCC	1.18 (1.14–1.22) **	1.15 (1.11–1.19) **
NE	1.76 (1.49–2.08) **	1.70 (1.41–2.03) **
Adenosquamous	1.56 (1.33–1.83) **	1.58 (1.33–1.87) **
GIST	0.26 (0.15–0.46) **	0.16 (0.07–0.36) **
**Tumor stage**		
Localized	1 (reference)	1 (reference)
Locally advanced	1.48 (1.42–1.54) **	1.78 (1.70–1.87) **
Distant	3.19 (3.06–3.32) **	4.13 (3.94–4.34) **
**Race**		
Non-Hispanic white	1 (reference)	1 (reference)
Non-Hispanic black	1.30 (1.24–1.37) **	1.31 (1.24–1.38) **
Hispanic	1.09 (1.03–1.15) **	1.11 (1.05–1.18) **
Other	1.03 (0.97–1.10)	1.04 (0.97–1.11)
**Living area**		
Counties in metropolitan areas of 1 million people	1 (reference)	1 (reference)
Counties in metropolitan areas of 250,000 to 1 million people	0.99 (0.96–1.03)	0.98 (0.95–1.02)
Counties in metropolitan areas of 250,000 people	1.12 (1.06–1.17) **	1.13 (1.07–1.19) **
Nonmetropolitan counties adjacent to a metropolitan area	1.06 (1.00–1.11)	1.07 (1.01–1.13) *
Nonmetropolitan counties not adjacent to a metropolitan area	1.08 (1.01–1.15)	1.09 (1.02–1.17) *
**Income per year**		
<$35,000	1 (reference)	1 (reference)
$35,000–44,999	0.98 (0.85–1.13)	1.04 (0.89–1.22)
$45,000–54,999	0.88 (0.77–1.01)	0.93 (0.80–1.08)
$55,000–64,999	0.88 (0.77–1.01)	0.92 (0.79–1.07)
$65,000–74,999	0.83 (0.72–0.95) **	0.86 (0.74–0.99) *
$75,000+	0.76 (0.67–0.87) **	0.80 (0.69–0.92) **
**Radiation**		
No	1 (reference)	1 (reference)
Yes	0.86 (0.83–0.89) **	0.86 (0.83–0.88) **
**Chemotherapy**		
No	1 (reference)	1 (reference)
Yes	0.83 (0.81–0.86) **	0.87 (0.84–0.90) **
**Surgery**		
No	1 (reference)	1 (reference)
Yes	0.30 (0.29–0.31) **	0.27 (0.25–0.28) **

* *p* < 0.05, ** *p* < 0.01.

**Table 3 medsci-12-00070-t003:** Multivariate Cox proportional hazard regression analyses of factors affecting all-cause mortality and cancer-related mortality among US patients diagnosed with esophageal cancer between 2010 and 2017.

Characteristics	Overall Mortality: Adjusted Proportional Hazard Ratio (95% Confidence Interval)	Cancer-Related Mortality: Adjusted Proportional Hazard Ratio (95% Confidence Interval)
**Gender**		
Female	1 (reference)	1 (reference)
Male	1.15 (1.11–1.20) **	1.13 (1.08–1.18) **
**Age at diagnosis, y.o.**		
00–39	1 (reference)	1 (reference)
40–59	1.21 (1.04–1.41) *	1.16 (1.00–1.36)
60–79	1.27 (1.09–1.48) **	1.15 (0.98–1.34)
80+	1.62 (1.39–1.90) **	1.40 (1.20–1.65) **
**Marital status**		
Married	1 (reference)	1 (reference)
Single	1.11 (1.07–1.15) **	1.09 (1.04–1.13) **
Divorced/separated	1.19 (1.14–1.25) **	1.20 (1.14–1.26) **
Widowed	1.16 (1.10–1.22) **	1.13 (1.07–1.20) **
**Tumor Subtype**		
Adenocarcinoma	1 (reference)	1 (reference)
SCC	1.38 (1.27–1.49) **	1.45 (1.33–1.59) **
NE	1.72 (0.99–2.97)	2.12 (1.17–3.85) *
Adenosquamous	2.06 (1.24–3.42) **	2.41 (1.37–4.26) **
GIST	0.29 (0.13–0.65) **	0.07 (0.01–0.52) **
**Race**		
Non-Hispanic white	1 (reference)	1 (reference)
Non-Hispanic black	1.09 (1.03–1.15) **	1.10 (1.03–1.16) **
Hispanic	0.98 (0.93–1.04)	0.99 (0.93–1.05)
Other	0.97 (0.91–1.04)	0.98 (0.91–1.05)
**Living area**		
Counties in metropolitan areas of 1 million people	1 (reference)	1 (reference)
Counties in metropolitan areas of 250,000 to 1 million people	0.98 (0.95–1.02)	0.97 (0.93–1.01)
Counties in metropolitan areas of 250,000 people	1.03 (0.97–1.09)	1.03 (0.97–1.10)
Nonmetropolitan counties adjacent to a metropolitan area	1.02 (0.96–1.09)	1.02 (0.96–1.09)
Nonmetropolitan counties not adjacent to a metropolitan area	1.04 (0.96–1.12)	1.05 (0.97–1.14)
**Income per year**		
<$35,000	1 (reference)	1 (reference)
$35,000–44,999	1.02 (0.88–1.18)	1.09 (0.93–1.27)
$45,000–54,999	0.94 (0.82–1.08)	0.99 (0.85–1.16)
$55,000–64,999	0.96 (0.83–1.11)	1.01 (0.87–1.19)
$65,000–74,999	0.89 (0.77–1.03)	0.93 (0.79–1.09)
$75,000+	0.85 (0.73–0.98) *	0.89 (0.76–1.05)
**Radiation**		
No	1 (reference)	1 (reference)
Yes	0.95 (0.92–0.98) **	0.94 (0.91–0.98) **
**Chemotherapy**		
No	1 (reference)	1 (reference)
Yes	0.52 (0.50–0.54) **	0.51 (0.49–0.53) **
**Surgery**		
No	1 (reference)	1 (reference)
Yes	0.37 (0.36–0.39) **	0.35 (0.34–0.37) **

* *p* < 0.05, ** *p* < 0.01.

**Table 4 medsci-12-00070-t004:** Multivariate Cox proportional hazard regression analyses of factors affecting all-cause mortality and esophageal cancer-related mortality among US patients between 2010 and 2017, taking into account the interaction between tumor stage and subtype.

Tumor Stage and Subtype (Subtype# Stage)	OM	CSM
AC Localized	1 (reference)	1 (reference)
AC locally advanced	2.15 (2.03–2.28) **	2.70 (2.52–2.88) **
AC distant	3.35 (3.17–3.55) **	4.36 (4.08–4.66) **
SCC locally advanced	0.64 (0.58–0.70) **	0.57 (0.51–0.63) **
SCC distant	0.73 (0.67–0.80) **	0.69 (0.62–0.76) **
NE locally advanced	0.58 (0.31–1.10)	0.43 (0.22–0.87) *
NE distant	0.91 (0.51–1.64)	0.67 (0.36–1.27)
ASC locally advanced	0.69 (0.39–1.23)	0.57 (0.30–1.09)
ASC distant	0.62 (0.36–1.08)	0.52 (0.28–0.96) *
GIST locally advanced	3.20 (0.65–15.90)	14.47 (1.31–159.87) *
GIST distant	1.22 (0.34–4.35)	4.25 (0.44–40.96)

AC—adenocarcinoma, SCC—squamous cell carcinoma, NE—Neuroendocrine tumor, ASC—adenosquamous, GIST—Gastrointestinal Stromal Tumor. * *p* < 0.05, ** *p* < 0.01.

## Data Availability

The data used and/or analyzed in this study are available in the Surveillance, Epidemiology, and End Results (SEER) Database of the National Cancer Institute (http://seer.cancer.gov, accessed on 6 June 2024).

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
