# Peer review of "The Impact of Tumor Stage and Histopathology on Survival Outcomes in Esophageal Cancer Patients over the Past Decade"

_medsci, 2024, doi:10.3390/medsci12040070_

Round 1
Reviewer 1 Report
Comments and Suggestions for Authors This study evaluated the impact of EC staging and histopathology on mortality rate. Numerous pieces of evidence have shown that staging and histopathology of EC are associated with mortality. This topic lacks novelty. The sample size and specific geographical location of the study may have certain significance. Limitation part should be listed seperately, and the content is too short. Introduction part is too long. Comments on the Quality of English LanguageMinor editing of English language required.
Author Response
This study evaluated the impact of EC staging and histopathology on mortality rate. Numerous pieces of evidence have shown that staging and histopathology of EC are associated with mortality. This topic lacks novelty. The sample size and specific geographical location of the study may have certain significance.
Re: Thank you for your feedback; however, the authors respectfully disagree, as current literature does not provide evidence on how the interaction between tumor stage and histology influences mortality in esophageal cancer.
Limitation part should be listed seperately, and the content is too short. Introduction part is too long.
Re: Thank you for your insights; the limitations have been expanded and organized into a separate section, with changes highlighted in red. The introduction could not be further condensed without compromising the intended message.
Reviewer 2 Report
Comments and Suggestions for Authors
The study is of significant importance and can be published in the journal. However, there are few suggestions that the authors needs to change before the article can be published:
1. The method section has too many sub-sections and sub-sub-sections which can be combined and streamlined to make the readers experience better.
2. Since most of the patient are male, and older aged patients, how did the authors avoid the bias? Did the authors consider weighted average method and if not why?
3. some details on how certain covariates were chosen for inclusion and how they impact the model’s sensitivity could enhance the study's transparency. Consider adding a sensitivity analysis or explaining why a p-value threshold of 0.01 was selected.
4. Your analysis includes various treatment modalities (radiation, chemotherapy, and surgery), providing a comprehensive view of their impact on mortality. Expanding on the limitations and potential confounding effects of treatment selection could help clarify whether the observed survival benefits are due to the treatment itself or associated patient characteristics
5. Expanding on the fourth point the authors could also include the recent use of ML/AI algorithms with the newer imaging modalities such as hyperspectral or multispectral imaging for esophageal cancer diagnosis and how it has helped endoscopists such as:
Chou, Chu-Kuang, Riya Karmakar, Yu-Ming Tsao, Lim Wei Jie, Arvind Mukundan, Chien-Wei Huang, Tsung-Hsien Chen, Chau-Yuan Ko, and Hsiang-Chen Wang. 2024. "Evaluation of Spectrum-Aided Visual Enhancer (SAVE) in Esophageal Cancer Detection Using YOLO Frameworks" Diagnostics 14, no. 11: 1129. https://doi.org/10.3390/diagnostics14111129
Fang, Yu-Jen, Chien-Wei Huang, Riya Karmakar, Arvind Mukundan, Yu-Ming Tsao, Kai-Yao Yang, and Hsiang-Chen Wang. 2024. "Assessment of Narrow-Band Imaging Algorithm for Video Capsule Endoscopy Based on Decorrelated Color Space for Esophageal Cancer: Part II, Detection and Classification of Esophageal Cancer" Cancers 16, no. 3: 572. https://doi.org/10.3390/cancers16030572
Wei-Chih Liao, Arvind Mukundan, Cleorita Sadiaza, Yu-Ming Tsao, Chien-Wei Huang, and Hsiang-Chen Wang, "Systematic meta-analysis of computer-aided detection to detect early esophageal cancer using hyperspectral imaging," Biomed. Opt. Express 14, 4383-4405 (2023)
Author Response
- The method section has too many sub-sections and sub-sub-sections which can be combined and streamlined to make the readers experience better.
Re: Thank you for this keen observation , the methods section has been made more concise and subsections have been combined to make the readers experience better and changes have been highlighted in red
- Since most of the patient are male, and older aged patients, how did the authors avoid the bias? Did the authors consider weighted average method and if not why?
Re: Thank you for acknowledging our study. We have included a multivariate analysis to minimize bias and confounding factors, which is a more robust approach compared to using a weighted average method.
- some details on how certain covariates were chosen for inclusion and how they impact the model’s sensitivity could enhance the study's transparency. Consider adding a sensitivity analysis or explaining why a p-value threshold of 0.01 was selected.
Re: Thank you for acknowledging our study and for your insightful observation. We retrieved data from the publicly available SEER database, from which we selected the covariates.
A p-value of less than 0.01 was chosen due to its statistical significance.
- Your analysis includes various treatment modalities (radiation, chemotherapy, and surgery), providing a comprehensive view of their impact on mortality. Expanding on the limitations and potential confounding effects of treatment selection could help clarify whether the observed survival benefits are due to the treatment itself or associated patient characteristics
Re: Thank you for your insightful observation. We conducted a multivariate analysis that accounted for confounding factors to eliminate any bias, confirming that the observed survival benefits are statistically attributable to the treatment itself.
- Expanding on the fourth point the authors could also include the recent use of ML/AI algorithms with the newer imaging modalities such as hyperspectral or multispectral imaging for esophageal cancer diagnosis and how it has helped endoscopists such as:
Chou, Chu-Kuang, Riya Karmakar, Yu-Ming Tsao, Lim Wei Jie, Arvind Mukundan, Chien-Wei Huang, Tsung-Hsien Chen, Chau-Yuan Ko, and Hsiang-Chen Wang. 2024. "Evaluation of Spectrum-Aided Visual Enhancer (SAVE) in Esophageal Cancer Detection Using YOLO Frameworks" Diagnostics 14, no. 11: 1129. https://doi.org/10.3390/diagnostics14111129
Fang, Yu-Jen, Chien-Wei Huang, Riya Karmakar, Arvind Mukundan, Yu-Ming Tsao, Kai-Yao Yang, and Hsiang-Chen Wang. 2024. "Assessment of Narrow-Band Imaging Algorithm for Video Capsule Endoscopy Based on Decorrelated Color Space for Esophageal Cancer: Part II, Detection and Classification of Esophageal Cancer" Cancers 16, no. 3: 572. https://doi.org/10.3390/cancers16030572
Wei-Chih Liao, Arvind Mukundan, Cleorita Sadiaza, Yu-Ming Tsao, Chien-Wei Huang, and Hsiang-Chen Wang, "Systematic meta-analysis of computer-aided detection to detect early esophageal cancer using hyperspectral imaging," Biomed. Opt. Express 14, 4383-4405 (2023)
Re; Thank you for this valuable point, the above references have been added to the study and highlighted in red in the discussion section
Reviewer 3 Report
Comments and Suggestions for Authors
Well-written and well-presented, this article highlights the fact that many risk factors for various histopathologies remain unrecognized. Importantly, it raises awareness about the significant role that lifestyle plays in the occurrence of EC.
Author Response
Well-written and well-presented, this article highlights the fact that many risk factors for various histopathologies remain unrecognized. Importantly, it raises awareness about the significant role that lifestyle plays in the occurrence of EC.
Re:Thank you very much for recognizing our study and appreciate your feedback back on our study to shed light on impact of lifestyle in EC.
Round 2
Reviewer 1 Report
Comments and Suggestions for Authors
This topic lacks novelty.
A conference abstract with similar content was published in the Journal of Clinical Oncology. However, several data in the published abstract are different. I'm not sure if these are two different studies or one study. If this is one study, why is the data different. The website for published abstracts is as follows: https://ascopubs.org/doi/10.1200/JCO.2024.42.16_suppl.e16054
Comments on the Quality of English LanguageModerate editing of English language required.
Author Response
We are reaching out regarding our study titled "The Impact of Tumor Stage and Histopathology on Survival Outcomes in Esophageal Cancer Patients Over the Past Decade," manuscript ID medsci-3253634. One of the reviewer comments pointed out that a conference abstract with similar content was published in the Journal of Clinical Oncology, highlighting some differences in the data. The reviewer is unsure if these represent two separate studies or a single study, and has requested clarification on the discrepancies.
To clarify, our study was initially submitted as an abstract based on data from patients treated between 2010 and 2017. The current manuscript, however, offers a more detailed analysis, incorporating updated data from the SEER database, which includes patients from 2010 to 2017. We have published multiple times with Medical Sciences, with our most recent paper focusing on CMML. While two reviewers have recognized the novelty of our study, one reviewer has questioned it. We feel that this criticism is unwarranted, as the interaction between tumor histology and tumor stage has not been thoroughly explored in the existing literature.
We would greatly appreciate your attention to this matter.
Thank you for your consideration.
Round 3
Reviewer 1 Report
Comments and Suggestions for Authors
None